# Molecular patterns identify distinct subclasses of myeloid neoplasia

Tariq Kewan [1,2,11] ✉, Arda Durmaz[1,3,11], Waled Bahaj[1], Carmelo Gurnari [1,4], Laila Terkawi[1], Hussein Awada [1], Olisaemeka D. Ogbue [1], Ramsha Ahmed[1], Simona Pagliuca[1,5], Hassan Awada[6], Yasuo Kubota[1], Minako Mori [1], Ben Ponvilawan[1], Bayan Al-Share[7], Bhumika J. Patel[1], Hetty E. Carraway [1], Jacob Scott[1,3], Suresh K. Balasubramanian[7], Taha Bat[8], Yazan Madanat [8], Mikkael A. Sekeres[9], Torsten Haferlach [10], Valeria Visconte [1,11] ✉ & Jaroslaw P. Maciejewski [1,11] ✉

Genomic mutations drive the pathogenesis of myelodysplastic syndromes and acute myeloid leukemia. While morphological and clinical features have dominated the classical criteria for diagnosis and classification, incorporation of molecular data can illuminate functional pathobiology. Here we show that unsupervised machine learning can identify functional objective molecular clusters, irrespective of anamnestic clinico-morphological features, despite the complexity of the molecular alterations in myeloid neoplasia. Our approach reflects disease evolution, informed classification, prognostication, and molecular interactions. We apply machine learning methods on 3588 patients with myelodysplastic syndromes and secondary acute myeloid leukemia to identify 14 molecularly distinct clusters. Remarkably, our model shows clinical implications in terms of overall survival and response to treatment even after adjusting to the molecular international prognostic scoring system (IPSS-M). In addition, the model is validated on an external cohort of 412 patients. Our subclassification model is available via a web-based open-access resource (https://drmz.shinyapps.io/mds_latent).

The myelodysplastic syndromes (MDS) are a collection of diseases encompassing a pathologically distinct, broad spectrum of myeloid disorders, some of which represent stages of the natural history of leukemia[1,2]. Until now, morphological features, later enhanced by cytogenetic abnormalities, have dominated the pathology criteria for MDS diagnoses. These can be limited by inter-observer variability, restricted resolution, and lack of functional correspondence to molecular underpinnings[3,4]. Widely-used MDS prognostic classification schemes may be convergent, as they group cases with similar features yet different molecular origins; or divergent, as they assign cases with similar molecular lesions into different pathomorphological sub-entities[5]. Moreover, when considering molecular features,

[1]Department of Translational Hematology and Oncology Research, Taussig Cancer Institute, Cleveland Clinic, Cleveland, OH, USA. [2]Department of Hematology and Medical Oncology, Yale University, New Haven, CT, USA. [3]Systems Biology and Bioinformatics Department, School of Medicine, Case Western Reserve University, Cleveland, OH, USA. [4]Department of Biomedicine and Prevention, Ph.D. in Immunology, Molecular Medicine and Applied Biotechnology, University of Rome Tor Vergata, Rome, Italy. [5]Department of Clinical Hematology, CHRU de Nancy, Nancy, France. [6]Roswell Park Comprehensive Cancer Center, Buffalo, NY, USA. [7]Department of Hematology and Oncology, Karmanos Cancer Institute, Wayne State University, Detroit, MI, USA. [8]Department of Internal Medicine, Division of Hematology and Oncology, University of Texas Southwestern Medical Center, Dallas, TX, USA. [9]Division of Hematology, Sylvester Cancer Center, University of Miami, Miami, FL, USA. [10]MLL Munich Leukemia Laboratory, Munich, Germany. [11]These authors contributed equally: Tariq Kewan, Arda Durmaz, Valeria Visconte, Jaroslaw P. Maciejewski. ✉e-mail: tariqkewan@gmail.com; visconv@ccf.org; maciejj@ccf.org

morphology-based classifications overemphasize specific parameters (e.g., blasts), which may represent essentially the stage of the disease, as opposed to molecular evolution. As a result, blast-defined MDS subtypes would contain a mixture of cases with various molecular derivations[5–9]. Classification schemes according to clinical features are more practical, but apart from the weight of cytogenetics and molecular mutations on prognosis[10], clinical prognostication does not reflect the disease pathogenesis[5,10–12].

The advent of next generation sequencing (NGS) has led to the discovery of a multitude of mutations in various genes and recognition of the tremendous molecular diversity and clonal hierarchy within myeloid malignancies[13–15]. These factors, along with cytogenetics, constitute the underpinnings of MDS pathogenesis. Given their complexity, attempts to consolidate mutational patterns into broader disease sub-entities have been made, with conventional integrative approaches of classical, clinical, and pathomorphological features used as a gold standard in supervised analytic strategies, including the new molecular international prognostic scoring system (IPSS-M)[10]. Consequently, the patterns of molecular features have been analyzed to fit into morphologic groups, with limited success given the complexity of mutations and their interactions, particularly with respect to disease progression[11,16]. Therefore, updated strategies may be needed to deconvolute this molecular diversity and generate subdivisions of patients with MDS whose disease fits within molecular pattern similarities, better reflecting prognosis and which could then be targeted with specialized therapeutic approaches. Machine learning (ML) analytic methods, as demonstrated in acute myeloid leukemia (AML)[17], provide new opportunities to integrate the molecular pathogenesis by identifying relevant patterns, which could serve as molecular sub-entities[11,16,18,19].

Here, we took advantage of a large, well-annotated cohort of patients with MDS and secondary AML (sAML) to test the hypothesis that related molecular patterns can be analyzed in an unbiased/unsupervised fashion to characterize molecularly defined configurations of MDS/sAML. We used a similarity-based ML approach to cluster patients into molecular subgroups, further validated based on clinical features.

## Results
### Unsupervised clustering of the molecular architecture of MDS and sAML reveals molecular subgroups regardless of histological or clinical features

Among the 3588 patients included in this cohort, 735 (20%) had sAML, 774 (22%) had higher-risk MDS (HR-MDS), and 2079 (58%) had lower-risk MDS (LR-MDS). Abnormal karyotype was found in 1548 cases (43%) (Table 1), and 2763 patients (77%) had at least one somatic mutation, with 284 cases (8%) harboring > 4 mutations (Supplementary Fig. 1).

Using unsupervised ML analysis of the mutational panel in our cohort, we identified 14 molecular clusters (MC1-MC14) according to distinct genomic features (Fig. 1A, B). The number of MCs was determined based on the highest silhouette value (Fig. 1A). The MCs size varied; for example, 26% of the cases were assigned to MC2 and only 2% to MC3 (Table 2). The most distinctive clinical and molecular features defining the MCs were identified (Table 2, Supplementary Fig. 2). Overall, the most important genomic features used in the model were quantified based on the mean decrease in accuracy (Supplementary Figs. 3 and 4).

Our ML model performance was then validated internally and externally. For the internal validation, we randomly selected training (80%, $n = 2870$) and test (20%, $n = 718$) sets for five-fold cross-validation to assess the fit of our model (See supplementary for details). Based on the highest silhouette value in each fold, the majority of the folds (3 out of 5) showed 14 clusters as optimal number of total clusters, similar to the full cohort, suggesting robust strategy for our approach (Supplementary Fig. 5A). Asymmetric and symmetric

calculation of Adjusted-Rand Index (ARI) between the folds showed a minimum ARI of 0.85 (Supplementary Fig. 5B, C). The external validation was conducted using an independent cohort of 412 MDS/sAML patients (Supplementary Tables 4, 5, and 6) with a different patient clinical composition distinct from the original cohort. Based on the mean decrease in accuracy, we selected and compared the most important characteristics between the original and the validation cohorts. As expected, no significant differences in these two cohorts in cytogenetics and molecular features in most MCs were observed (Fig. 1C). However, the assignment to clusters was different given the significant variations in the baseline features of the validation cohort (Supplementary Tables 4). Furthermore, we have used a Bayesian latent class analysis as a baseline model for comparison using

**Table 1 | Clinical, cytogenetic, and molecular characteristics of myelodysplastic and secondary acute myeloid leukemia cohorts at baseline**

| Variables | All | MDS | | sAML |
|---|---|---|---|---|
| | | Lower-risk | Higher-risk | |
| Total population | 3588 | 2079 | 774 | 735 |
| Test cohort (%) | 718 (20) | 426 (20) | 156 (20) | 136 (19) |
| Training cohort (%) | 2870 (80) | 1653 (80) | 618 (80) | 599 (82) |
| Age, median (IQR) | 72 (64–77) | 72 (64–78) | 72 (65–78) | 71 (63–76) |
| Gender | | | | |
| Male, n (%) | 2143 (60) | 1211 (58) | 478 (62) | 454 (62) |
| Female, n (%) | 1444 (40) | 867 (42) | 296 (38) | 281 (38) |
| Labs | | | | |
| WBC ($10^9$/L), median (IQR) | 4 (3–11) | 5 (3–9) | 4 (2–8) | 5 (2–20) |
| Hemoglobin (g/dL), median (IQR) | 10 (9–11) | 10 (9–11) | 10 (9–11) | 9 (8–10) |
| Platelet ($10^9$/L), median (IQR) | 112 (50–240) | 160 (78–305) | 98 (48–177) | 50 (24–91) |
| BM blast %, median (IQR) | 4 (2–13) | 2 (1–3) | 9 (7–13) | 37 (25–61) |
| Diagnosis | | | | |
| sAML[a] | 735 (20) | — | — | 735 (100) |
| MDS-SLD[b] | 513 (14) | 513 (25) | — | — |
| MDS-MLD[b] | 776 (22) | 776 (37) | — | — |
| 5q syndrome | 165 (5) | 165 (8) | — | — |
| MDS/MPN | 363 (10) | 363 (17) | — | — |
| CMML 1/2 | 246 (7) | 189 (9) | 57 (7) | — |
| MDS-EB1 | 369 (10) | — | 369 (48) | — |
| MDS-EB2 | 348 (10) | — | 348 (45) | — |
| Others[c] | 73 (2) | 73 (4) | — | — |
| Cytogenetics | | | | |
| Normal | 2023 (57) | 1259 (61) | 446 (58) | 318 (43) |
| Abnormal | 1548 (43) | 807 (39) | 327 (42) | 414 (57) |
| Number of MT | | | | |
| 0 | 825 (23) | 491 (24) | 110 (14) | 224 (30) |
| 1–2 | 1666 (46) | 1068 (51) | 351 (46) | 247 (33) |
| 3–4 | 813 (23) | 416 (21) | 233 (30) | 163 (22) |
| >4 | 284 (8) | 104 (5) | 80 (10) | 101 (14) |

[a] sAML is defined as acute myeloid leukemia evolving from MDS or MDS/MPN, or diagnosed in the setting of MDS-related cytogenetic abnormalities. [b] MDS-SLD-RS and MDS-MLD-RS are included in this group. [c] MDS/MPN-RS-T and MDS-U are included in this group. *IQR* interquartile range, *WBC* white blood cell count, *MDS* myelodysplastic syndrome, *sAML* secondary acute myeloid leukemia, *BM* bone marrow, *SLD* single lineage dysplasia, *MLD* multi-lineage dysplasia, *CMML* chronic myelomonocytic leukemia, *MPN* myeloproliferative neoplasm, *EB* excess blast, *MT* mutations. No statistical comparison was performed.

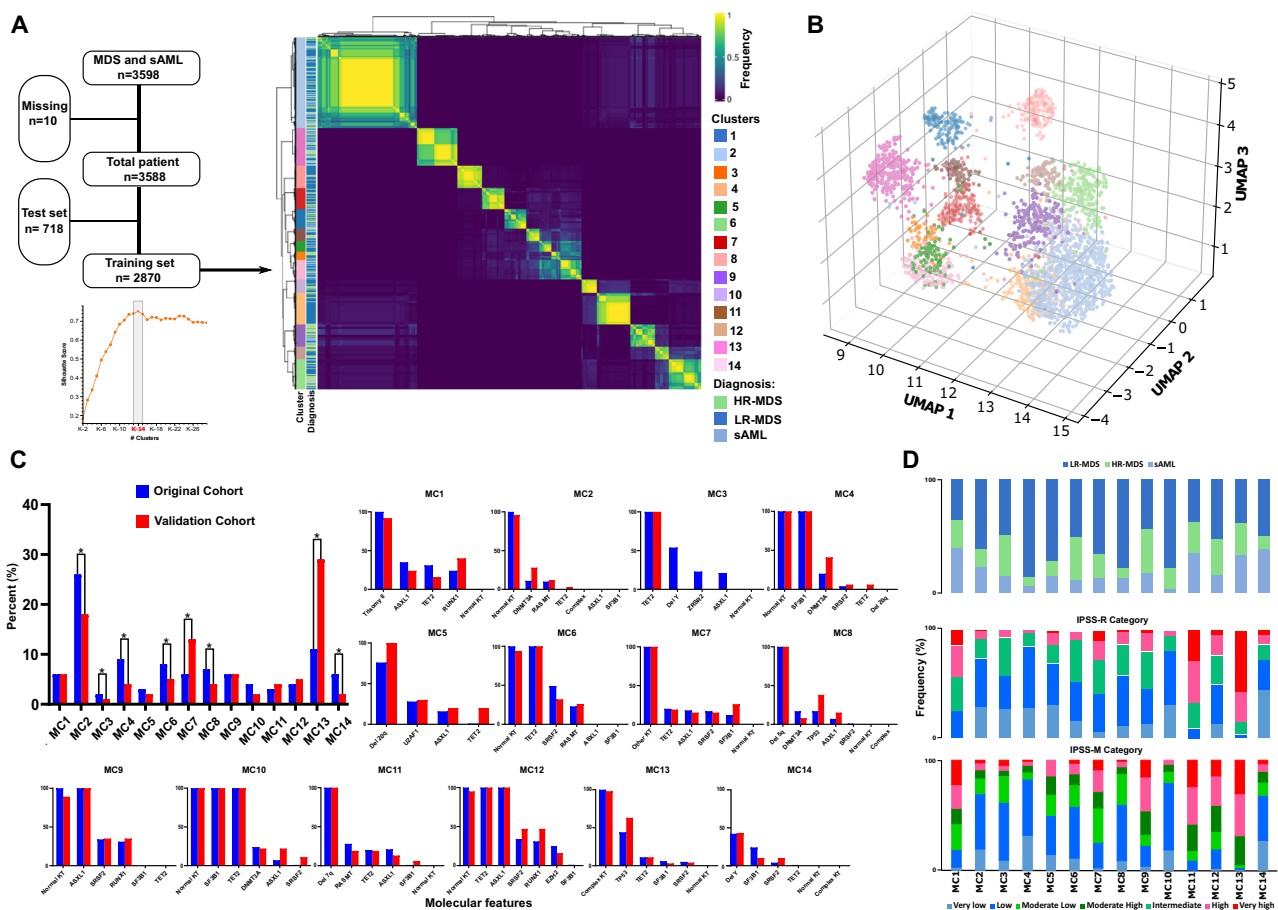

**Fig. 1 | Genomic clusters of myelodysplastic syndromes and secondary acute myeloid leukemia identified by unsupervised analysis. A** Unsupervised clustering of binary mutation profiles performed through iteratively clustering subsamples of original data and keeping track of the frequency of pairwise co-occurrence of samples in the same cluster. HR high-risk, LR low-risk, sAML secondary acute myeloid leukemia. **B** To visualize the clusters in a three-dimensional space, we have generated an exemplary dimension-reducing space using UMAP coupled with the autoencoder model. A 16-dimensional linear embedding of binary mutation profiles was generated and reduced to 3d using UMAP in a nonlinear fashion. The legend color is consistent with each genomic cluster in panel **A**. Uniform Manifold Approximation and Projection (UMAP). **C** Bar graph illustrating the frequency of each genomic cluster in the original and the validation cohort. Significant differences between frequencies based on the two-sided Chi-square test are indicated by asterisks, $P$ value $\leq 0.05$ was considered statistically significant. No significant differences in the frequency of the molecular features were found between the original and the validation cohorts within patients assigned to similar molecular clusters (MCs). Graphs from MC1-MC14 illustrate the frequency of the most important molecular features in the original and the validation cohorts. KT karyotype, Del deletion. **D** upper panel: Bar graph showing the relative frequency of low-risk myelodysplastic syndrome (LR-MDS), high-risk myelodysplastic syndrome (HR-MDS), and secondary acute myeloid leukemia (sAML) for each molecular cluster (MC). The middle panel is showing the relative frequency of different Revised International Prognostic Scoring System (IPSS-R) among different clusters. The lower panel is showing the relative frequency of different Molecular International Prognostic Scoring System (IPSS-M) among different clusters.

R package BayesLCA[20]. As expected, clustering using the Bayesian approach resulted in more granular and lower resolution clusters where substantial overlap with proposed risk groups was present. Nevertheless, the proposed Autoencoder-based clustering was able to further stratify the BayesLCA-based clusters (Supplementary Fig. 6).

**Molecular clusters composition and phenotype associations**
The composition of the MCs was clinically distinct, reflecting differing morphological diagnoses and bone marrow (BM) blast counts (Table 2, Supplementary Fig. 7A–C). For instance, the highest rate of female patients was observed in MC8 (70%). Patients assigned to MC2 and MC7 were younger (median age: 69 and 68 years, respectively). LR-MDS patients comprised most of MC8 (78%), MC10 (78%), and MC5 (72%). In addition, more than 50% of the cases within MC2, MC4, MC6, MC7, and MC14 were LR-MDS patients. Conversely, HR-MDS and sAML cases comprised more than 30% of MC3, MC6, MC9, and MC12. MDS single lineage dysplasia (MDS-SLD) compromised 41% of MC4 and 26% of MC10. MDS excess blast (EB) 1/2 constituted more than 30% of MC3,

MC6, and MC9 (Supplementary Fig. 7B). The majority of CMML cases clustered into MC12, MC6, and MC3. In addition, MCs demonstrated distinct clinical differences within laboratory values (Supplementary Fig. 7D–F). For instance, patients in MC1, MC11, and MC13 had significantly lower platelet counts (median of 87, 48, and 76 109/L, respectively, $p$-value $< 0.001$) compared to other clusters. The highest median hemoglobin level (11 g/dL) was observed in patients assigned to MC6, whereas patients within MC1, MC8, MC11, and MC13 had lower values (median around 9 g/dL).

When we applied reverse analysis, the majority of the sAML cases populated MC2 (28%), MC13 (18%), MC1 (11%), and MC14 (10%). HR-MDS cases were mainly classified in MC2 (19%), MC6 (15%), MC13 (14%), and MC9 (11%). Finally, LR-MDS clustered in MC2 (27%) and MC4 (13%; Supplementary Fig. 8). Moreover, the distribution of different revised international scoring system (IPSS-R) and molecular international prognostic scoring system (IPSS-M) risk groups among our MCs were distinct and heterogenous (Fig. 1D). Although most of the cases included in MC11 and MC13 were very-high and high-risk groups

**Table 2 | Clinical, cytogenetic, and molecular characteristics of all clusters**

| Variables | All | MC1 | MC2 | MC3 | MC4 | MC5 | MC6 | MC7 | MC8 | MC9 | MC10 | MC11 | MC12 | MC13 | MC14 | P-value |
|---|---|---|---|---|---|---|---|---|---|---|---|---|---|---|---|---|
| Total population | 3588 | 201 | 920 | 76 | 313 | 107 | 301 | 225 | 236 | 219 | 143 | 121 | 130 | 391 | 205 | |
| Test cohort | 718 (20) | 39 (19) | 178 (19) | 12 (16) | 58 (19) | 18 (17) | 58 (19) | 54 (24) | 55 (23) | 37 (17) | 31 (22) | 20 (17) | 26 (20) | 87 (22) | 45 (22) | |
| Training cohort | 2870 (80) | 162 (80) | 742 (81) | 64 (84) | 255 (82) | 89 (83) | 243 (81) | 171 (76) | 181 (77) | 182 (83) | 112 (78) | 101 (84) | 104 (80) | 304 (78) | 160 (78) | |
| Age, median (IQR) | 72 (64–77) | 74 (67–80) | 69 (59–76) | 75 (69–76) | 72 (66–78) | 72 (65–78) | 74 (68–78) | 68 (59–74) | 74 (68–80) | 72 (64–77) | 74 (70–78) | 70 (62–75) | 75 (69–80) | 71 (63–77) | 71 (63–77) | <0.001 |
| Gender | | | | | | | | | | | | | | | | <0.001 |
| Male | 2143 (60) | 131 (65) | 519 (56) | 60 (79) | 182 (58) | 72 (67) | 203 (67) | 135 (60) | 71 (30) | 163 (74) | 76 (53) | 70 (58) | 91 (70) | 217 (56) | 153 (75) | |
| Female | 1444 (40) | 70 (35) | 401 (44) | 16 (21) | 131 (42) | 35 (33) | 98 (33) | 90 (40) | 165 (70) | 56 (26) | 67 (47) | 51 (42) | 39 (30) | 174 (45) | 51 (25) | |
| Labs | | | | | | | | | | | | | | | | |
| WBC (10$^9$/L) | 4 (3–11) | 6 (3–14) | 5 (3–12) | 7 (4–11) | 6 (4–9) | 5 (3–10) | 5 (3–13) | 5 (3–12) | 4 (3–6) | 6 (3–22) | 5 (4–7) | 4 (3–19) | 10 (3–25) | 3 (2–8) | 6 (3–13) | <0.001 |
| Hb (g/dL) | 10 (9–11) | 9 (8–11) | 10 (9–12) | 10 (9–12) | 10 (9–11) | 10 (9–11) | 11 (10–13) | 10 (9–11) | 9 (8–10) | 10 (9–11) | 10 (9–11) | 9 (8–10) | 10 (9–12) | 9 (8–10) | 10 (8–11) | <0.001 |
| Platelet (10$^9$/L) | 112 (50–240) | 87 (32–163) | 104 (54–221) | 118 (68–160) | 307 (184–420) | 92 (39–171) | 100 (66–154) | 111 (46–191) | 217 (77–321) | 96 (38–220) | 232 (146–333) | 48 (24–131) | 83 (47–131) | 76 (30–102) | 109 (40–235) | <0.001 |
| BM blast % | 4 (2–13) | 12 (4–37) | 4 (1–14) | 9 (3–12) | 2 (1–4) | 3 (1–6) | 6 (3–12) | 3 (1–7) | 4 (2–5) | 7 (3–14) | 2 (1–4) | 8 (4–24) | 8 (3–14) | 12 (3–22) | 5 (1–29) | <0.001 |
| Diagnosis | | | | | | | | | | | | | | | | <0.001 |
| LR-MDS | 2079 (58) | 73 (36) | 566 (62) | 37 (49) | 268 (86) | 77 (72) | 152 (51) | 147 (65) | 184 (78) | 96 (44) | 112 (78) | 46 (38) | 68 (52) | 150 (38) | 103 (50) | |
| HR-MDS | 774 (22) | 50 (25) | 148 (16) | 28 (37) | 27 (9) | 14 (13) | 116 (39) | 49 (22) | 21 (9) | 86 (39) | 26 (18) | 33 (27) | 42 (32) | 111 (28) | 23 (11) | |
| s-AML[a] | 735 (21) | 78 (39) | 206 (22) | 11 (15) | 18 (6) | 16 (15) | 33 (11) | 29 (13) | 31 (13) | 37 (17) | 5 (4) | 42 (35) | 20 (15) | 130 (33) | 79 (39) | |
| Diagnosis, % | | | | | | | | | | | | | | | | |
| MDS-SLD[b] | 14 | 8 | 16 | 16 | 41 | 19 | 6 | 14 | 0.4 | 6 | 26 | 3 | 5 | 10 | 19 | <0.001 |
| MDS-MLD[b] | 22 | 12 | 22 | 16 | 38 | 35 | 19 | 22 | 7 | 19 | 50 | 12 | 18 | 17 | 19 | <0.001 |
| 5q syndrome | 5 | 0 | 0.1 | 0 | 0 | 0 | 0 | 0.4 | 67 | 0 | 0 | 0 | 1 | 1 | 0 | <0.001 |
| MDS/MPN | 10 | 11 | 18 | 7 | 5 | 18 | 5 | 14 | 2 | 10 | 2 | 8 | 6 | 6 | 9 | <0.001 |
| CMML 1/2 | 7 | 6 | 4 | 13 | 2 | 1 | 24 | 11 | 1 | 12 | 1 | 9 | 25 | 1 | 2 | <0.001 |
| MDS-EB1 | 10 | 10 | 10 | 18 | 6 | 8 | 19 | 9 | 4 | 19 | 13 | 15 | 8 | 9 | 7 | <0.001 |
| MDS-EB2 | 10 | 13 | 6 | 15 | 2 | 5 | 15 | 10 | 4 | 16 | 4 | 10 | 22 | 19 | 4 | <0.001 |
| Others[c] | 2 | 1 | 2 | 1 | 1 | 1 | 1 | 7 | 0.4 | 1 | 0 | 9 | 1 | 4 | 2 | <0.001 |
| Cytogenetics | | | | | | | | | | | | | | | | |
| Normal | 2023 (57) | 1 (1) | 920 (100) | 0 (0) | 313 (100) | 106 (100) | 300 (99.7) | 0 (0) | 0 (0) | 217 (99) | 142 (99) | 0 (0) | 130 (100) | 0 (0) | 0 (0) | |
| Abnormal | 1548 (43) | 200 (100) | 0 (0) | 76 (100) | 0 (0) | 0 (0) | 1 (0.3) | 221 (100) | 236 (100) | 2 (1) | 1 (1) | 121 (100) | 0 (0) | 391 (100) | 205 (100) | |
| Number of MT | | | | | | | | | | | | | | | | |
| 0 | 825 (23) | 40 (20) | 446 (49) | 0 (0) | 0 (0) | 22 (21) | 0 (0) | 39 (17) | 75 (32) | 0 (0) | 0 (0) | 34 (28) | 0 (0) | 92 (24) | 77 (38) | |
| 1–2 | 1666 (46) | 70 (35) | 366 (40) | 45 (59) | 235 (75) | 55 (51) | 162 (54) | 109 (48) | 130 (55) | 57 (26) | 63 (44) | 54 (36) | 7 (5) | 219 (56) | 104 (51) | |
| 3–4 | 813 (23) | 57 (28) | 77 (8) | 26 (34) | 68 (22) | 21 (20) | 117 (39) | 50 (22) | 26 (11) | 114 (52) | 70 (49) | 32 (26) | 71 (55) | 64 (16) | 20 (10) | |
| >4 | 284 (8) | 34 (17) | 31 (3) | 5 (7) | 10 (13) | 9 (8) | 22 (7) | 27 (12) | 5 (2) | 48 (22) | 10 (7) | 1 (1) | 52 (40) | 16 (4) | 4 (2) | |

[a] sAML is defined as acute myeloid leukemia evolving from MDS or MDS/MPN, or diagnosed in the setting of MDS related cytogenetic abnormalities. High risk MDS (HR-MDS) was defined based on BM blast <5%. Low risk MDS (LR-MDS) was defined based on BM blast <5%. [b] MDS-SLD-RS and MDS-MLD-RS are included in this group. [c] MDS/MPN-RS-T and MDS-U are included in this group. IQR interquartile range, WBC white blood cell count, MDS myelodysplastic syndrome, sAML secondary acute myeloid leukemia, BM bone marrow, SLD single lineage dysplasia, MLD multi-lineage dysplasia, CMML chronic myelomonocytic leukemia, MPN myeloproliferative neoplasm, EB excess blast, MT mutations. Two-sided nonparametric Wilcoxon matched-pairs signed rank test and Chi square test were used to compare differences across groups for numerical and categorical variables respectively. P value ≤ 0.05 was considered statistically significant.

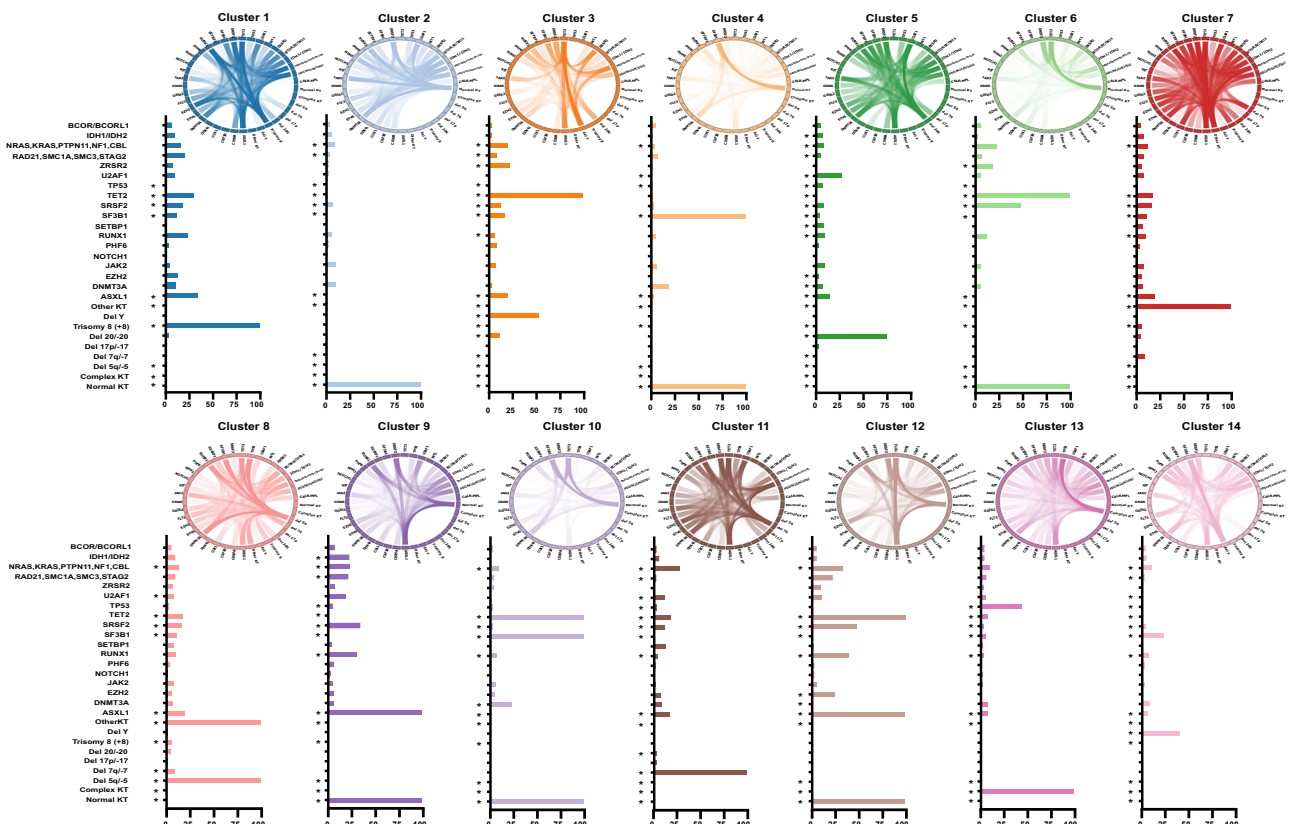

**Fig. 2 | Genomic features drive each genomic group.** Bar plots represent the mutational profiles of all genomic clusters (clusters 1 to 14) and their importance. Asterisks denote the most important genomic features based on an importance cutoff of a mean decrease in accuracy ≥ 0.01. The circos diagrams above each cluster show the pairwise co-occurrence of mutations in all clusters. The colors of circos diagrams correspond to the clusters. The percentage of mutational co-occurrence between first and second gene mutations is represented by the color intensity of the ribbon connecting both genes.

according to IPSS-R, both clusters continue to contain patients from other risk groups who share the same molecular configuration. Similarly, very high risk and high risk IPSS-M groups were mainly enriched within MC1, MC9, MC11, MC12, and MC13 (Fig. 1D, lower panel).

Blast percentages in MCs were consistent with the risk distribution of cases, and the median blast percentage was consistent with the composition of each MC (Table 2 and Supplementary Fig. 9A). For instance, while MC1 and MC13 had a median blast percentage of >10%, MC2 and MC4 had a median of <5%, consistent with the enrichment of early-stage (LR-MDS) cases within the latter MCs. Overall, MC1 and MC13 had significantly higher odds for ≥20% BM blast percentage while MC2 and MC4 had higher odds for <20% BM blast percentage (Supplementary Fig. 9B).

### Machine learning-derived clusters reflect functional relationships

Broad cluster-specific analyses revealed that MC4 cases all had NK and *SF3B1* mutations (Fig. 2). Similarly, all MC10 cases had NK and *SF3B1* mutations in addition to *TET2* mutations (100%). *DNMT3A* mutations were present in 20 and 24% of MC4 and MC10, respectively. MC2, MC6, and MC8 demonstrated distinct genomic signatures: MC2 included cases with NK only (100%) and some *DNMT3A* (11%), *JAK2* (11%), and *RAS* pathway (10%) mutations; MC6 cases had similar features to MC2 but were also enriched in *SRSF2* (49%) and *RAS* mutations (23%); MC8 was characterized by the presence of del5q/-5 (100%), *DNMT3A* (17%), and *TP53* (17%) mutations. MC3 included cases with *TET2* (100%), *ZRSR2* (23%), and *ASXL1* (21%) mutations with delY (54%). MC14 included cases with delY (42%) but without *TET2* mutations, distinct from MC3. In contrast, MC12 included cases with *TET2* (100%), *ASXL1* (100%), *SRSF2*

(48%), *RUNX1* mutations (40%), and NK (100%) similar to MC9, which contained *ASXL1* (100%) with *SRSF2* (34%) and *RUNX1* (31%) but lacked *TET2* mutations. While both MC10 and MC12 seemed similar regarding features such as *TET2* mutations and normal cytogenetics, there were differences based on the absence and/ or presence of certain somatic mutations and cytogenetic abnormalities. The frequency of *SF3B1* mutations was higher in MC10, while MC12 was enriched with *ASXL1* mutations (Fig. 2 and Supplementary Fig. 4). Finally, only 7% of patients in MC10 had more than 4 concurrent somatic mutations, a feature characterizing up to 40% of patients in MC12.

MC5 grouped cases with del20q/-20 (76%) and *U2AF1* mutations (28%). MC7 was characterized by other cytogenetic abnormalities, not including del5q/-5 compared to MC8. MC1 was characterized by trisomy 8 (100%), *ASXL1* (35%), *TET2* (31%), and *RUNX1* (24%) mutations. MC11 included cases with del7q/-7 (100%) and *RAS* pathway mutations (28%). Finally, MC13 contained cases with complex karyotype (100%) and *TP53* (44%) mutations. The resultant MC signatures are illustrated in Supplementary Table 7. To understand the frequency of each mutation within the identified clusters, we also show the distribution of each genomic mutation and cytogenetic abnormalities across clusters (Supplementary Fig. 10). Based on the molecular associations of each group (Fig. 2), we illustrated that patients with same mutation/ cytogenetic abnormalities may be assigned to different MCs based on the other associated co-existing lesions.

### MDS molecular clusters have clinical correlates

We explored differences in overall survival (OS) across the identified MCs (Fig. 3A, B). As expected, the high degree of molecular heterogeneity translated to divergent survival in each subset (Supplementary

Fig. 11). However, median survival profiles failed to overlap with the external validation cohort, possibly due to the low number of cases assigned to individual clusters (Supplementary Table 8). By grouping MCs according to OS, we distinguished 5 risk categories (Fig. 3C and Supplementary Table 9), which were recapitulated in the external validation cohort in terms of significant separation of the survival curves between risk groups based on the Kaplan-Meier (KM) estimates and the global log-rank test statistic (Fig. 3D). The KM curves for the external validation and training sets showed no difference except for MC13 where comparison of MC3,5,8,10,14 was hampered by low number of classified cases (Supplementary Fig. 12, Supplementary Table 8). In order to further alleviate this issue, we compared the aggregated risk groups using CoxPH as well which showed similar HR profiles with the training dataset albeit with decreased statistical significance. This was due to the limitation in suboptimal aggregation of MCs (Supplementary Table 10). We also detected discrete survival differences in patients ($N$ = 2863) treated with hypomethylating agents (HMAs) and/or allogeneic hematopoietic stem cell transplant. Even after accounting for the different treatments, the risk groups continued to show significant differences in OS using semi-parametric Cox-PH model, where nonparametric alternative random-forest survival model showed improved separation in untreated cohort compared to treated samples as well (Supplementary Fig. 13A, B). Interestingly, we noticed association between treatment responses and our MCs. For instance, a higher response rate to HMAs (according to the International Working Group criteria[21]) occurred in patients assigned to MC9, MC10, and MC12 (36%, 33%, and 32%, respectively), of which 29 and 71% of the treated cases had complete remission (CR)/ marrow complete remission (mCR) and hematological improvement (HI), respectively. In contrast, response rates in patients assigned to MC1, MC13, MC3, and MC7 were 13%, 13%, 14%, and 15%, respectively; (Fig. 3E, F), of which CR and HI were achieved in 23 and 27% of the treated cases, respectively, and none of the patients achieved mCR. Venetoclax was used concurrently with HMAs in 6 and 7% of the patients in MC11 and MC13. Multivariate logistic regression analysis showed that MC9 (odd ratio [OR]: 2.2, 95%CI: 1.2-3.9) and MC13 (OR: 0.6, 95%CI: 0.4–0.9) were associated with significantly different HMAs response rates.

This ML method focused on clustering patients with molecular similarities. The blast percentage within MCs did not appear to affect survival after 25 months. For instance, although MC13 contained 38 and 33% of LR and HR-MDS patients, respectively, the prognosis was homogenously worse when compared to other MCs. Using Cox-Proportional Hazard model accounting for relevant clinical variables including age, gender, BM blast percent, HMAs treatment, and allogeneic HSCT, the assigned risk groups based on our clustering showed significant survival differences (Supplementary Fig. 13B). Compared to the LR Group-1 (OS [95% CI]; 93 months [42–132]), patients within Group-5 (OS [95% CI]; 9 [4–24]), Group-4 (OS [95% CI]; 17 [5–53]), Group-3 (OS [95% CI] 33 [12–92]), and Group-2 (OS [95% CI]; 62 [19–188]) had significantly worse OS.

Our clustering model was able to highlight the significant survival differences among patients assigned to the similar IPSS-R risk group but to different MCs (Supplementary Fig. 14). For instance, we observed significant differences in OS among patients assigned to very low risk IPSS-R based on our MCs (HR:1.9, 95%CI: 1.5–2.8). Patients assigned to low, moderate low, high and very high IPSS-M risk groups had survival differences based on our model (Supplementary Fig. 15). To rule out possible confounding effect of IPSS-M and IPSS-R in estimating the OS differences among our risk groups, we generated a Cox PH model including IPSS-M/IPSS-R and clinical variables (Supplementary Tables 11 and 12). As expected, both models continued to show significant association with OS. In order to compare the utility of the proposed model in time-to-event modeling with IPSS-M using OS, we bootstrapped Harrel's C-index differences. Incorporating clinical variables (age, sex, blast percent, hemoglobin, and platelet counts) showed no significant difference between IPSS-M and the proposed clusters (Supplementary Fig. 16). Overall, our model was comparable and significantly overlapping with IPSS-M in which high risk MCs/ groups had higher IPSS-M scores (Supplementary Fig. 17).

## Discussion

While MDS classification schemes evolved as useful clinical diagnostic or prognostic tools, diagnostic criteria according to genomic signatures reflective of molecular pathogenesis have not been established[1,11,22]. Furthermore, previous attempts to incorporate mutations into prognostic schemes to increase their predictive precision resulted in considering only a handful of consequential mutations[17]. One of the reasons for the notorious inability to establish reproducible genotype/phenotype associations might be the application of primarily supervised strategies using traditional statistics and clinical classifications (reliant on subjective nosology and time-dependent parameters) as a gold standard. The recent IPSS-M[10] mitigates some of these issues but it still remains most heavily dependent on clinical parameters which represent stage of the disease rather than molecular pathogenesis[10]. Indeed, the tremendous diversity and complexity of molecular lesions hamper the application of conventional bioanalytic methods.

To overcome the limitations of these traditional approaches, our study applied modern ML strategies to objectively integrate molecular features able to decipher patient sub-cohorts with known and/or previously cryptic associations. This strategy was not meant to compete with or replace current well-established prognostication tools[5,10] but rather illuminate the genetic sub-classification of MDS and related conditions in an operator-independent fashion according to molecular correlations and mutual functional proximity. Despite the exclusion of anamnestic clinical criteria, the resultant scheme yielded a reproducible and validated system of genetically related disease clusters reflective of the genomic pathogenesis and prognosis, irrespective of established standards. Notably, our molecular classification has enabled the recognition of cases with convergent molecular mechanisms, *e.g.*, for the rational selection of suitable therapies. Moreover, the personalized risk stratification method is independent of disease duration and stage. It does not involve blast count, whose predictive weight dominates most of the older disease schemes and the recently proposed molecular prognostication model[1,10,11].

Our ML-based molecular model defines unique clusters according to the previously described genomic features and their combinations known to influence MDS and sAML outcomes[2,11,23–25]. Moreover, the analysis of the invariant cluster-defining molecular combinations points towards relationships or convergent pathways. For instance, even historically well-defined subgroups such as MDS with deletion 5q, were included along with sAML cases within MC8, indicating that heterogeneous morphological subtypes can be grouped within the same cluster based on shared molecular lesions explaining common pathogenesis and disease behavior. Illustrative examples of such molecular associations are presented in the supplementary materials (Supplementary notes).

With the exception of *TP53* and *SF3B1* mutations, the recent IPSS-M assigns fixed scores for patients sharing certain mutations. However, as we illustrated in our MCs, not all patients with similar mutations will have similar pathophysiology and co-existing mutations. We demonstrated that patients with splicing factor mutations are assigned to different subgroups based on the presence of other epigenetic modulator mutations. Similarly, our results confirmed the functional distinction between monoallelic *TP53* (mainly in MC8) and biallelic *TP53* mutations (MC13). Recent studies have suggested a functional association between *U2AF1* mutations and RNA-splicing genes located on chromosome 20 (*e.g.*, *GNAS*), this link was illustrated in MC5 in which *U2AF1* mutations is associated with del20q[26].

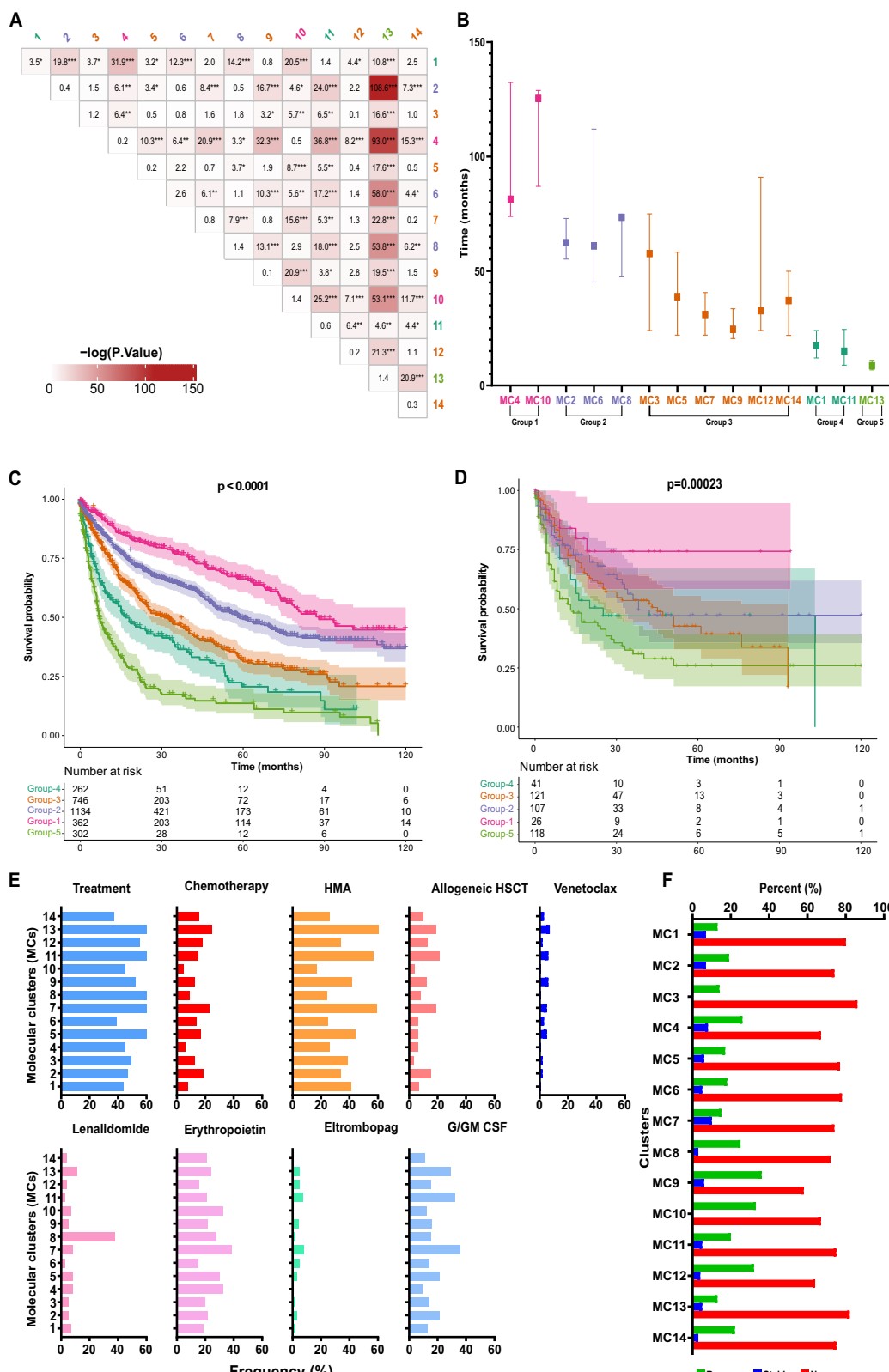

**Fig. 3 | Survival outcomes and model validation. A** Pairwise survival comparison between the identified genomic clusters using log-rank test. Asterisks indicate the significant -log (P-values) with increasing significance values; 0.05, 0.01, 0.001 respectively. **B** Median overall survival in months with 95% confidence interval (CI) of all molecular clusters and assigned risk groups based on time-to-event profiles. *N* = 3588 patients. **C** Kaplan-Meier survival curves of all risk groups in the original cohort with the associated 95% CI. **D** Kaplan-Meier survival curves of all risk groups in the external validation cohort with the associated 95% CI. **E** Bar graph showing the frequency of various first-line treatments used in each cluster. HMA hypomethylating agents, HSCT hematopoietic stem cell transplantation, G/GM CSF granulocyte/monocyte colony-stimulating factor. **F** Histogram bars represent the response to hypomethylating agents treatment among different clusters (MC) based on the international working group criteria[21].

Unlike previous prognostication systems highly dependent on the blast count[5], our MCs were heterogeneous in this regard. This observation raises many questions about the implication of BM blast percentages on molecularly-based diagnoses. Indeed, our ML-based scheme indicates that BM blast may correlate more with the stage of the disease rather than the molecular architecture. For instance, although MC13 included patients with the worst prognosis, almost 1/3 of the cases in this cluster had low blast counts while sharing a similar molecular makeup with sAML, reflecting different stages of the same disease. Analogous observations were made in other clusters containing molecularly similar patients at various points of their clinical course. Significant survival associations with BM blasts and MCs also suggest that these variables capture different information regarding the disease pathogenesis. It is important to emphasize that the recent attempts to integrate cyto-molecular features into MDS classification for personalized approaches were also based on traditional clinical parameters, which always outweighed the variables derived from the genomic makeup. For instance, when analyzing the fraction of explained variation attributable to different prognostic factors for OS, BM blast percentage, age, and sex alone accounted for more than 50%. In comparison, molecular features only had limited power in the proposed model (-30%)[11].

BM blast percentage was associated with OS in a time-dependent manner. For instance, an arbitrary cutoff of 25 months showed a significant difference in blast count effects on OS. While high blast percentage is associated with worse OS and LFS according to IPSS-R and IPSS-M prognostication models[5,10], the effects of differential blast counts are less eminent in patients already characterized by high-risk features (complex karyotype, *TP53* mutations >3 mutations, etc.)[16]. Using our molecular approach, a heterogeneous number of blasts was observed within individual MCs, indicating that cases with similar molecular make-up can present at different stages of the disease in which patients with higher blast percent represent a more advanced disease phase. Irrespective of the prognostic value of blasts, our study, because of its blast-agnostic approach, provides the advantage of identification of high-risk cases early in the disease course to implement the most appropriate therapies. Moreover, the distribution of prognostic groups (IPSS-R and IPSS-M) within MC (Fig. 1D) is consistent with the theory that distinct outcomes may result from similar molecular genotypes due to diverse treatments, comorbidities, age, and other factors.

In our model, focusing on the objective molecular signature to characterize the features of different clusters with the exclusion of morphological and clinical data may seem a limitation. However, we believe that clinical and morphological features constitute the results of genetic hits. We showed that our molecular clustering of MDS successfully identified unique patterns of genomic associations and possibly uniform/similar pathogenesis even if individual connections cannot be rationally discerned on this junction. We acknowledge that additional parameters such as allelic configuration/burden, mutation types, clonal/subclonal burden, and germline predisposition may add a significant value if incorporated, perhaps helping to further substratify some of the more heterogeneous clusters. In addition, our NGS panel included mutations in 40 genes (uniformly tested), and some excluded molecular mutations, although rare, may have important clinical and pathogenesis impacts. Another limitation of any analytic strategy (supervised/unsupervised) is that less common mutations remain unappreciated because of the lack of statistical power. This is also a flaw of our approach, which we attempted to mitigate by combining mutations affecting the same functional pathways and identifying rare hits confined strictly to one cluster to allow for inferences in terms of their functional associations.

In conclusion, despite the complexity and the diversity of molecular alterations in MDS and sAML, by deploying artificial intelligence analytics, we were able to discern functional and pathologically related, objective clusters irrespective of the anamnestic clinico-morphological features. The purpose of such classification is to identify patients with common features possibly suggesting a similar rate of sensitivity to targeted agents, and for the clinical investigators to rationally test efficacy of drugs in these molecularly-related sub-entities. The scoring systems benchmarked solely on survival may bundle patients with unrelated pathophysiology into the same groups, which if used for treatment indications (low risk disease vs. high risk disease) may result in heterogeneous response patterns. Molecular clustering would help to avoid such a scenario. Our model is available as an open-access resource for clinicians and researchers to establish relationships between molecular profiles irrespective of the stage of disease for rational selection of therapies.

## Methods

### Patient cohort

We assembled a large cohort of patients diagnosed with MDS, MDS/myeloproliferative neoplasm (MPN) including chronic myelomonocytic leukemia (CMML), and sAML to generate a comprehensive genomic data set. Patient data from the Cleveland Clinic ([CC], $n = 1627$), The Munich Leukemia Laboratory ([MLL], $n = 1275$), and publicly available data sets (The BEAT AML master trial and The EuroMDS cohort Patients, $n = 686$)[11,27] were combined to form a cohort of 3588 MDS and sAML patients (Supplementary Tables 1 and 2). Targeted NGS results at the time of diagnosis were collected and adjusted to analyze the most common somatic myeloid mutations (Supplementary Table 3). Electronic medical records were reviewed to collect clinical parameters at the time of diagnosis and from resources accessible online from the publicly shared data sources (EuroMDS). Samples were collected after obtaining written informed consent according to the protocols approved by the respective institutional review boards (see Supplementary Materials). The study was approved by the Cleveland Clinic IRB.

### Genetic studies

For the data collected at CC, paired tumor and germline DNAs were used for whole exome sequencing (WES)[28–30]. Data were validated using a TruSeq Custom Amplicon Kit (Illumina) (Supplementary Table 3). Variants were annotated using Annovar and filtered using an in-house bioanalytic pipeline[15,28,30,31]. The gene sequencing methods of publicly shared MDS and sAML patients can be found in the original articles[11,27]. For validation, an independent cohort of MDS/sAML patients (UT Southwestern medical center and Karmanos Cancer Institute was used; see Supplementary Table 1 & Supplementary Methods).

### Statistical analyses

Our ML strategy was based on a consensus-clustering approach via autoencoders coupled with Gaussian-mixture modeling (GMM)[32]. The resultant model was validated internally and externally on an independent cohort (detailed description in the Supplementary Materials). In order to aggregate the MCs into distinct risk categories, we selected 5 risk groups within which MCs showed similar time-to-event profiles at t where $S(t) > 0.25$ for all MCs where 0.25 is chosen arbitrarily. Although this subjective strategy is not optimal, stratification made OS amenable to succinct investigation. Our genomic subclassification model is available via web-based open-access resources as well (https://drmz.shinyapps.io/mds_latent).

### Reporting summary

Further information on research design is available in the Nature Portfolio Reporting Summary linked to this article.

## Data availability

All raw and processed data used to generate the results of this study is provided, processed and raw data with scripts to ensure

reproducibility can be found at https://github.com/ardadurmaz/mds_latent. Raw DNA sequencing is provided in the database of Genotypes and Phenotypes (dbGaP) under the NCBI under accession number phs001898.v1.p1. Genomic data is available through the dbGAP-controlled access database [https://www.ncbi.nlm.nih.gov/projects/gap/cgi-bin/study.cgi?study_id=phs001898.v1.p1]. Access can be granted through dbGAP, and contact can be made to Jaroslaw P. Maciejewski (maciejj@ccf.org). There are no restrictions on who will be granted access. A list of samples included in this parent phs is included at https://github.com/ardadurmaz/mds_latent. Sequencing data for all other patients in our study was done as part of the NGS diagnostic test. All other information is provided in the Supplementary Information/ Tables.

## Code availability

The scripts used for unsupervised clustering and figure generation are deposited to https://github.com/ardadurmaz/mds_latent and is publicly available (https://doi.org/10.5281/zenodo.7757422). The available web access allows users to select molecular and cytogenetic features (named genomic profile) and obtain survival probability and probability of enrichment of such features in each cluster.

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

## Acknowledgements

We thank our sources of funding: the HENRY & MARILYN TAUB FOUNDATION (J.P.M.), grants R01HL118281 (to J.P.M.), R01HL123904 (to J.P.M., R.A.P.), R01HL132071 (to J.P.M., R.A.P.), R35HL135795 (all to J.P.M), AA&MDSIF (to V.V., S.P., J.P.M), VeloSano Pilot Award, Vera and Joseph Dresner Foundation–MDS (to V.V.). C.G. was supported by a grant from the Edward P. Evans Foundation. This work used the High-Performance Computing Resource in the Core Facility for Advanced Research Computing at Case Western Reserve University.

## Author contributions

T.K. collected and analyzed clinical and molecular data, interpreted results, and wrote the manuscript. A.D. applied ML-based methods, analyzed the data, interpreted the results, and wrote the manuscript. T.K. and A.D. equally contributed as first authors. W.B., C.G., S.P., Hussein.A., Hassan.A. analyzed data, provided important insights into the manuscript, and edited the manuscript. W.B. and C.G. equally contributed as second authors. O.D.O, R.A, L.T, and B.S. collected data and edited the manuscript. J.S., B.J.P., H.E.C., and Y.K. provided feedback on manuscript preparation and writing. T.H., Y.M., S.K.B, T.B., M.A.S, provided data and important insights to the manuscript. V.V., J.P.M. provided invaluable help to the manuscript preparation, generated and conceived the study design, designed figures and tables, and wrote the manuscript. V.V. and J.P.M. equally contributed as senior authors and corresponding authors. All authors participated in the critical review of the final paper and submission.

## Competing interests

Y.M. has received honoraria/consulting fees from BluePrint Medicines, GERON, OncLive and MD Education; participated in advisory boards and received honoraria from Sierra Oncology, Stemline Therapeutics, Blueprint Medicines, Morphosys, Taiho Oncology, Rigel Pharmaceuticals and Novartis; and received travel reimbursementfrom Blueprint Medicines, MD Education, and Morphosys. None of these relationships were related to this work. The remaining authors declare no competing interests.
