## [Peer Review File · Nature Communications]

Molecular patterns identify distinct subclasses of myeloid neoplasiaREVIEWER COMMENTS

Reviewer #1 (Remarks to the Author): expertise in AML genomics

In the current study, the authors utilized an unsupervised clustering approach based on autoencoders coupled with Gaussian-mixture modeling (GMM), which is one of the best methods for unsupervised clustering. This GMM method could rationally divide more than 3000 MDS and MPN patients into 14 clusters based on specific genetic alterations. Patients from each cluster also showed distinct clinical characteristics and prognosis. Further, the authors divided the 14 groups of patients into five combined risk groups according to their survival information: low, int-low, int-high, high and very high. This new risk stratification was further validated by an external patient cohort. Finally, the authors built a website for predicting the clusters of patients from the real world. Overall, there is a big patient cohort included into this study (over 3000 patients) and therefore the conclusions are relatively convincing. Although this research is of clinical interest and suggests a potentially novel risk stratification system for MDS and MPN patients, I have several major concerns regarding applicability of the presented data:

Major:

1. Table 2 reflects the similarities and differences of various clinical parameters in the 14 clusters. For some factors, such as platelet and diagnosis, distinct characteristics could be observed. For example, cluster 4 showed the highest platelet count and cluster 11 showed the lowest level. Cluster 6 and Cluster 12 showed the highest proportion of CMML as compared to other clusters. These data are interesting but not clearly visualized or emphasized based on the current version. If these data could be displayed with a figure and analyzed by statistics, it could further help readers understand the similarities and differences in the clinical parameters among 14 clusters.
2. Although in Figure 3B-D the authors divide the 14 clusters of 2806 patients into five combined risk groups, this novel risk stratification system is not used in the web tool. In the current version, the web tool is based on 14 individual clusters instead of risk stratification groups, which is less suitable for the clinical use. Preferably, the web tool could be organized similarly to IPSS-M calculator, that is, after entering the cytogenetic and molecular information of a patient, the corresponding risk stratification can be directly calculated rather than genetic clusters.
3. The clusters in the web tool are numbered in a different way when compared to the manuscript. For example, if we use Normal Karyotype and SF3B1 mutations as an input, the tool shows distribution probability for clusters 0-13, whereas cluster numbers 1-14 are used in the manuscript. Moreover, described above input settings result in the highest probability for cluster 13 identified in the manuscript by TP53 mutations and Complex Karyotype. The authors should correct discrepancies between clustering in the manuscript and online tool.
4. The current IPSS-M risk stratification contains molecular genetic information on 31 genes, which show the strongest prognostic significance in MDS patients. Therefore, the reviewer feels that it is important to show the proportions of different IPSS-M risk groups among 14 clusters or 5 prognostic risk stratifications as presented in Figure 1C, rather than only being limited to IPSS-R. Based on this comparison, the similarity and difference between the two models can be further explored. Furthermore, the current study would be of higher scientific and potential clinical value if the authors quantitatively compare the performance of their own model (reduced to 5 prognostic risk stratification groups) and the IPSS-M model. For example, receiver operating characteristic (ROC) curve, Calibration curve and Decision curve analysis (DCA) could be performed.
5. The stratification system containing 14 clusters is not designed to compete with the current IPSS-R or IPSS-M system as stated in the discussion. How will it be implemented in the MDS risk stratification in the clinical settings? In which situations can this system be applied?
6. Some of the clusters generated by the authors in the study are characterized by similar genetic signatures. For example, clusters 10 and 12 are characterized by Normal Karyotype and TET2 mutations with 100% frequency. If a random patient with Normal Karyotype and TET2 mutation without additional genetic lesions is entered into this system, to which cluster will it be assigned?

Importantly, cluster 10 is grouped into low risk group, whereas cluster 12 is assigned to Int-High risk group. Unfortunately, the current version of the web tool does not provide an answer to this question. This also raises the question of whether additional information, including blast % and data on cytopenias, are required in these cases for more precise stratification.

7. The authors mentioned that their clustering model was able to highlight survival differences among IPSS-R risk groups (Supplementary Figure 11). However, these survival differences are subtle. Moreover, groups with higher risk (groups 5 and 2) show superior survival inside low-IPSS-R and intermediate IPSS-R risk groups. Visually, some survival curves suffer from low sample size. Therefore, the information about patient numbers for each group and numbers at risk are needed for this Figure.

8. In Figure 3E, the authors compared the proportion of treatments in the 14 clusters. It would be very interesting if the authors could further compare the response of patients to different treatments in each cluster.

Minor:

1. In tables 1 and 2, the authors should describe how they identified low(er)-risk and high(er)-risk MDS patients. Was it done based on IPSS-R risk stratification? In this case, in which category were intermediate risk patients assigned?

2. Did the authors perform statistical analysis for MC1-14 shown in Figure 1D? For example, the data suggest that there could be differences for MC8 and MC12 in original and validation cohorts.

3. Please assign group numbers in Figure 3B and Supplementary Table 8 (e.g. low risk = 1, very high risk =5)

4. It is not completely clear what is shown in Supplementary Table 7. Are those data from the validation cohort and why median survival times are not available for many clusters? It would be helpful to provide patient numbers (n) for each cluster and present the data for training set in parallel.

5. It was also not clear to the reviewer why HR values for groups 2 and 3 in comparison to group 1 (low risk) are below 1 (Supplementary Table 9). There are also very big CIs despite very low p-values even for training set.

6. Some figures are not presented in the order described in the text. For example, the authors described Figure 2 instead of Figure 1D. Similarly, Supplementary Figure 12 was firstly described rather than Supplementary Figure 10. Supplementary Figure 13 was not described in the text at all.

Reviewer #2 (Remarks to the Author): expert in machine learning and biostatistics

The author proposed and validated a machine-learning model to classify large cohorts of MDS samples with cytogenetic and molecular features. The 14 molecular clusters identified in the manuscript have unique pathobiological associations, treatment responses, and prognoses. Specifically, MCs were split into five risk groups with different overall survival. The work brought genomic signatures to the attention of MDS classification.

Major:

1. Please provide more details about how the model is developed and evaluated in the supplementary methods section, including the following information:

What is the binary mutation profile? is it a variant or gene-level profile? How is it derived from the WES data? And should it be defined clearly in the manuscript?

How was the RF model trained? It is claimed in the Supplementary Methods section that the input is the binary mutation profile. However, features like Normal KT, Complex KT, Other KT were seen in many feature importance plots, such as Supplementary Figure 3 and 4.

2. A baseline model could be provided and compared with the model proposed. This will show the necessity of using this complex model involving autoencoder, GMM, hierarchical clustering, and RF.

3. With the current model evaluation results, I'm not convinced about the robustness.

The similar number of clusters in each fold (in Line 151) is not enough to show the robustness.

The calculation of the ARI is not clearly described in the manuscript or supplementary materials.

According to the R script named by cluster_folds.R in the GitHub page provided by the author, ARI here is not fair to show the consistency of results.

Suppose the internal data is divided into 5 sets: set 1, 2, 3, 4, and 5.

Fold-1 whole training process involves set 2, 3, 4, and 5;

Fold-2 whole training process involves set 1, 3, 4, and 5.

When calculating the ARI between Fold-1 and Fold-2:

$$[[ARI]]_{1,2} = ARI(\{ \text{labels of set 2,3,4,5 predicted by Fold1} \}, \{ \text{labels of set 2,3,4,5 predicted by Fold2} \})$$

But set 3, 4, and 5 were training data in both Fold-1 and 2. The cross-validation evaluation results shouldn't contain shared training samples. Thus, I suggest the author to calculate the ARI between Fold-1 and Fold-2 by:

$$[[ARI]]_{1,2} = ARI(\{ \text{labels of set 1 predicted by Fold1} \}, \{ \text{labels of set 1 predicted by Fold2} \})$$

What is the statistical test for "no significant differences" in Line 156?

4. Please provide summary figure or statistics for the claim in Line 171 "Blast percentages in MCs were consistent with the risk distribution of cases, and the median blast percentage was consistent with the composition of each MC." The current results provided are not clear for this conclusion.

5. Besides Supplementary Figure 11 and the example in the very low IPSS-R group, please provide more evidence to rule out the confounding factor effect of IPSS-R in studying the overall survival differences among MCs, since it was mentioned in Line 167 that "the distribution of different revised international prognostic scoring system (IPSS-R) risk groups among our MCs were distinct and heterogenous." This is essential in determining the novelty of the findings in OS differences.
Minor:

1. Figure 2 is rarely discussed in the manuscript. Should the author attach more writing to the results if it is essential, or put it in supplementary figures and take other plots to the main figure?

2. The color palette consistency

In Figure 1, please make sure each label has only one color. E.g., A and C have different label colors for LR-MDS, HR-MDS, and sAML.

The five risk group labels are inconsistent between Supplementary Figure 10A and others.

RESPONSE TO REVIEWERS' COMMENTS

Reviewer #1 (Remarks to the Author): expertise in AML genomics

Major:

1. Table 2 reflects the similarities and differences of various clinical parameters in the 14 clusters. For some factors, such as platelet and diagnosis, distinct characteristics could be observed. For example, cluster 4 showed the highest platelet count and cluster 11 showed the lowest level. Cluster 6 and Cluster12 showed the highest proportion of CMML as compared to other clusters. These data are interesting but not clearly visualized or emphasized based on the current version. If these data could be displayed with a figure and analyzed by statistics, it could further help readers understand the similarities and differences in the clinical parameters among 14 clusters.

Reply: We thank the reviewer for this valuable comment. To further clarify the significant differences in clinical parameters, including diagnosis and laboratory values, between the molecular clusters, we performed a statistical analysis and compared the clinical parameters between all clusters. We now added *P*-values to **Table-2**. To provide an additional visual representation of the comparisons of clinical factors across clusters, as suggested, we added a new **Supplementary Figure 7**. It is important to mention that MC6 (Normal karyotype, *TET2^{MT}* and *SRSF2^{MT}* [45%]) and MC12 (Normal karyotype, *TET2^{MT}*, *ASXL1^{MT}*, and *SRSF2^{MT}* [43%]) contained the highest proportion of patients with CMML but they have different molecular makeup. In addition, we emphasized the clinical differences between clusters in the main text, see below:

*"The majority of CMML cases clustered into MC12, MC6, and MC3. In addition, MCs demonstrated distinct clinical differences within laboratory values (**Supplementary Figure 7D-F**). For instance, patients in MC1, MC11, and MC13 had significantly lower platelet counts (median of 87, 48, and 76 $10^9/L$, respectively, *p*-value <0.001) compared to other clusters. The highest median hemoglobin level (11 g/dL) was observed in patients assigned to MC6, whereas patients within MC1, MC8, MC11, and MC13 had lower values (median around 9 g/dL)."*

Supplementary Figure-7:

2. Although in Figure 3B-D the authors divide the 14 clusters of 2806 patients into five combined risk groups, this novel risk stratification system is not used in the web tool. In the current version, the web tool is based on 14 individual clusters instead of risk stratification groups, which is less suitable for the clinical use. Preferably, the web tool could be organized similarly to IPSS-M calculator, that is, after entering the cytogenetic and molecular information of a patient, the corresponding risk stratification can be directly calculated rather than genetic clusters.

Reply: We want to thank the reviewer for this suggestion. We have now updated the web tool to include the risk stratification with the molecular clusters. We have also included

additional details in the supplementary material explaining the risk-model generation for the web tool. Specifically, we have included in the supplementary methods the following paragraph:

"In order to include the risk stratification to the assigned molecular clusters, we generated a random-forest classifier using the identified risk groups. as well. We then followed a similar approach for hyperparameter tuning. Furthermore, our risk stratification grouping has been added to the web tool to help with the clinical application of our model."

3. The clusters in the web tool are numbered in a different way when compared to the manuscript. For example, if we use Normal Karyotype and SF3B1 mutations as an input, the tool shows distribution probability for clusters 0-13, whereas cluster numbers 1-14 are used in the manuscript. Moreover, described above input settings result in the highest probability for cluster 13 identified in the manuscript by TP53 mutations and Complex Karyotype. The authors should correct discrepancies between clustering in the manuscript and online tool.

Reply: We want to thank the reviewer for pointing out the inconsistency between the web tool and the cluster stratification. Indeed, we have observed a mapping issue between the feature names, which has been fixed now. Cluster names (0-13) have now been updated to (1-14) and are consistent through the manuscript.

4. The current IPSS-M risk stratification contains molecular genetic information on 31 genes, which show the strongest prognostic significance in MDS patients. Therefore, the reviewer feels that it is important to show the proportions of different IPSS-M risk groups among 14 clusters or 5 prognostic risk stratifications as presented in Figure 1C, rather than only being limited to IPSS-R. Based on this comparison, the similarity and difference between the two models can be further explored. Furthermore, the current study would be of higher scientific and potential clinical value if the authors quantitatively compare the performance of their own model (reduced to 5 prognostic risk stratification groups) and the IPSS-M model. For example, receiver operating characteristic (ROC) curve, Calibration curve and Decision curve analysis (DCA) could be performed.

Reply: We thank the reviewer for this relevant and excellent comment. Based on the reviewer's suggestion, we calculated IPSS-M scores for the patients in our cohort (and added explanations as to that in the supplementary methods) and presented the frequencies of IPSS-M risk groups among our molecular clusters and our risk groups (**Figure 1D**, lower panel). As we emphasized in the article, our aim was not to provide a prognostic score, but to clarify the possible molecular interactions between different molecular mutations in patients assigned to similar molecular clusters, which for sure will affect prognosis, treatment response, and other clinical/prognostic features. However, to ensure that our machine learning model is applicable clinically and is not contradicting the current well-established molecular/clinical tools, we compared our model with IPSS-M using Harrel's C index (New **Supplementary Figures 15 and 16**). We also incorporated clinical

features (age, sex, blast percent, hemoglobin and platelet count) into our model and compared it with IPSS-M. This was added to the main text and the supplementary materials:

“Similarly, very high risk and high risk IPSS-M groups were mainly enriched within MC1, MC9, MC11, MC12, and MC13 (Figure 1D, lower panel).”

AND

“Patients assigned to low, moderate low, high and very high IPSS-M risk groups had survival differences based on our model (Supplementary Figure 15).”

AND

“In order to compare the utility of the proposed model in time-to-event modeling with IPSS-M using OS, we bootstrapped Harrel's C-index differences. Incorporating clinical variables (age, sex, blast percent, hemoglobin and platelet counts) showed no significant difference between IPSS-M and the proposed clusters (Supplementary Figure 16). Overall, our novel model was comparable and significantly overlapping with IPSS-M in which high risk MCs/groups had higher IPSS-M scores (Supplementary Figure 17).”

Figure 1D:

Supplementary Figure 15

Supplementary Figure 16

Supplementary Figure 17

5. The stratification system containing 14 clusters is not designed to compete with the current IPSS-R or IPSS-M system as stated in the discussion. How will it be implemented in the MDS risk stratification in the clinical settings? In which situations can this system be applied?

Reply: We thank the reviewer for the comment. As mentioned in the previous point, our aim was not to provide a prognostic score, but to clarify the possible molecular interactions

between different mutations in patients assigned to similar molecular clusters which for sure will affect prognosis, treatment response and other clinical/prognostic features. Our molecular-based approach was efficient in showing differences among patients with similar single mutations. The purpose of such molecular classification is to identify patients with possibly similar sensitivity to targeted agents, and for the clinical investigators to rationally test efficacy of novel drugs in molecularly related sub-entities. The scoring systems benchmarked on survival may group patients with unrelated pathophysiology into the same groups, which if used for indication for specific treatment (low risk disease vs. high-risk disease), may result in heterogeneous response patterns. The molecular clustering would help to avoid such a scenario. We added these points to our discussion:

"The purpose of such classification is to identify patients with common features possibly suggesting a similar rate of sensitivity to targeted agents, and for the clinical investigators to rationally test efficacy of novel drugs in these molecularly-related sub entities. The scoring systems benchmarked solely on survival may bundle patients with unrelated pathophysiology into same groups, which if used for treatment indications (low risk disease vs. high risk disease) may result in heterogeneous response patterns. The molecular clustering would help to avoid such a scenario."

6. Some of the clusters generated by the authors in the study are characterized by similar genetic signatures. For example, clusters 10 and 12 are characterized by Normal Karyotype and TET2 mutations with 100% frequency. If a random patient with Normal Karyotype and TET2 mutation without additional genetic lesions is entered into this system, to which cluster will it be assigned? Importantly, cluster 10 is grouped into low-risk group, whereas cluster 12 is assigned to Int-High risk group. Unfortunately, the current version of the web tool does not provide an answer to this question. This also raises the question of whether additional information, including blast % and data on cytopenias, are required in these cases for more precise stratification.

Reply: We thank the reviewer for this great point. We included the following explanation *"While both MC10 and MC12 seemed similar regarding features such as TET2^{MT} and normal cytogenetics, there were differences based on the absence and/ or presence of certain somatic mutations and cytogenetic abnormalities. The frequency of SF3B1^{MT} was higher in MC10, while MC12 was enriched with ASXL1^{MT} (Figure 2 and Supplementary Figure 4). Finally, only 7% of patients in MC10 had more than 4 concurrent somatic mutations, a feature characterizing up to 40% of patients in MC12"*.

In order to translate these complex interaction patterns, we also updated the version of the web tool as mentioned before to present the risk groups as well. We pointed out the molecular signatures of the clusters to make our model more applicable clinically but the importance of each feature in every single cluster identification is highlighted in **Figure 2** and **Supplementary Figure 4**.

7. The authors mentioned that their clustering model was able to highlight survival differences among IPSS-R risk groups (**Supplementary Figure 11**). However, these survival differences are subtle. Moreover, groups with higher risk (groups 5 and 2) show superior survival inside low-IPSS-R and intermediate IPSS-R risk groups. Visually, some survival curves suffer from low sample size. Therefore, information about patient numbers for each group and numbers at risk are needed for this Figure.

Reply: We thank the reviewer for this comment. We added the risk tables as required and updated the figure as needed (**Supplementary Figure 14**). Overall, our model did not contradict IPSS-R or IPSS-M, while it focused solely on robust molecular and cytogenetic features. Clinical features, mainly blasts percent, have a significant impact on survival that can mask and underestimate the effect of somatic mutations at the later stages of the disease. In addition to updating the requested numbers in the Figure, we also clarified our points in the discussion section:

*“Moreover, the distribution of prognostic groups (IPSS-R and IPSS-M) within MC (**Figure 1D**) is consistent with the theory that distinct outcomes may result from similar molecular genotypes due to diverse treatments, comorbidities, age and other factors.”*

8. In Figure 3E, the authors compared the proportion of treatments in the 14 clusters. It would be very interesting if the authors could further compare the response of patients to different treatments in each cluster.

Reply: We thank the reviewer for this clinically relevant comment. We chose response to treatment as one of the features that can illustrate the biological and clinical differences between the molecular clusters (cluster to cluster differences). We only assessed response to HMAs as shown in Figure 3 since it was the most used treatment in our cohort. Chemotherapy and allogenic HSCT were less observed in our cohort, hence precluding any further analysis because of lack of statistical power.

Minor:

1. In tables 1 and 2, the authors should describe how they identified low(er)-risk and high(er)-risk MDS patients. Was it done based on IPSS-R risk stratification? In this case, in which category were intermediate risk patients assigned?

Reply: We thank the reviewer for raising this point. We classified patients into high-risk MDS (HR-MDS) and low-risk MDS (LR-MDS) groups based on the bone marrow blast percent (5% cutoff). We clarified this point in the table legend:

"High risk MDS (HR-MDS) was defined based on $\geq 5\%$ bone marrow (BM) blast. Low risk MDS (LR-MDS) was defined based on BM blast $< 5\%$."

2. Did the authors perform statistical analysis for MC1-14 shown in Figure 1D? For example, the data suggest that there could be differences for MC8 and MC12 in original and validation cohorts.

Reply: We thank the reviewer for the comment. Statistical analysis was done to compare the molecular features between our cohort and validation cohorts. No significant differences were found. We clarified this point in the legend of **Figure 1**:

"Significant differences between frequencies based on the Chi-square test are indicated by asterisks. No significant differences in the frequency of the molecular features were found between the original and the validation cohorts within patients assigned to similar MCs."

3. Please assign group numbers in Figure 3B and Supplementary Table 8 (e.g. low risk = 1, very high risk =5)

Reply: We thank the reviewer for the comment. Risk group numbers were edited in both **Figure 3B** and **Supplementary Table 8**.

4. It is not completely clear what is shown in Supplementary Table 7. Are those data from the validation cohort and why median survival times are not available for many clusters? It would be helpful to provide patient numbers (n) for each cluster and present the data for training set in parallel.

Reply: We thank the reviewer for the comment. **Supplementary Table 7** shows the median OS for the validation cohort stratified by the molecular clusters 1-14. We added the total number in each cluster to the table. The median survival was not reported for some clusters due to the small number of patients.

Supplementary Table 7. Median overall survival with 95% Confidence Intervals for the external validation cohort

Cluster	Total number	Median Survival
Cluster-1	25	16 (12-NA)
Cluster-2	74	128 (38-NA)
Cluster-3	2	5.5 (0-NA)
Cluster-4	17	NA (19-NA)
Cluster-5	10	NA (30-NA)
Cluster-6	19	30 (20-NA)
Cluster-7	54	47 (21-NA)
Cluster-8	13	36 (32-NA)
Cluster-9	26	26 (12-NA)
Cluster-10	9	NA (NA-NA)
Cluster-11	16	25 (13-NA)
Cluster-12	19	51 (11-NA)
Cluster-13	117	13 (8-28)
Cluster-14	10	NA (12-NA)

5. It was also not clear to the reviewer why HR values for groups 2 and 3 in comparison to group 1 (low risk) are below 1 (Supplementary Table 9). There are also very big CIs despite very low p-values even for training set.

Reply: We thank the reviewer for this comment. In the previous analysis, we reported the hazard ratios (HR) using log scale, we edited the table and reported the exponential scale to avoid confusion. There was a mistake regarding the upper CI as well and has been updated. We updated the results in **Supplementary Table 9**.

Risk Group	Training Set (HR, CI, P-Value)	Validation Set (HR, CI, P-Value)
Group-2	1.41 (1.15- 1.74, .001)	1.87 (0.79-4.24, 0.152)
Group-3	2.27 (1.84-2.81, <0.0001)	2.12 (0.91-4.92, 0.080)
Group-4	3.49 (2.72-4.48, <0.0001)	2.52 (1.01-6.27, 0.047)
Group-5	5.69 (4.50-7.19, <0.0001)	3.76 (1.63-8.69, 0.001)

6. Some figures are not presented in the order described in the text. For example, the authors described Figure 2 instead of Figure 1D. Similarly, Supplementary Figure 12 was firstly described rather than Supplementary Figure 10. Supplementary Figure 13 was not described in the text at all.

Reply: We thank the reviewer for this comment. We reviewed the order of appearance of the figures and ensured that all main and supplementary figures are referred to in the main text in the correct sequence.

Reviewer #2 (Remarks to the Author): expert in machine learning and biostatistics
Major:

1. Please provide more details about how the model is developed and evaluated in the supplementary methods section, including the following information.

What is the binary mutation profile? is it a variant or gene-level profile? How is it derived from the WES data? And should it be defined clearly in the manuscript?

Reply: We want to thank the reviewer for the comment.

The binary mutation profile indicates presence or absence of mutations (meaning at a gene-level) irrespective to variant allele frequency. Supplemental material contains information pertinent to the sequencing techniques and references to the original studies (Supplementary Methods, Pages 31-32). We included:

"Specifically, mutations and cytogenetic alterations were encoded as binary features based on presence or absence."

How was the RF model trained? It is claimed in the Supplementary Methods section that the input is the binary mutation profile. However, features like Normal KT, Complex KT, Other KT were seen in many feature importance plots, such as Supplementary Figure 3 and 4.

Reply: Further clarifications are made in the supplementary document. Specifically, we have trained the RF model using all the features as binary input regardless of whether the feature represents large genomic alterations or mutations aggregated (mutated/non-mutated) to the gene level, optimizing the Gini coefficient as a target function. Cytogenetic features (Normal KT, Complex KT, Other KT) are important genetic alterations and were included as binary input (presence/absence) for the RF model training. Specifically, we have included as follows:

"Specifically, mutations and cytogenetic alterations were encoded as binary features based on presence or absence."

Furthermore, we have included another section in the supplementary document detailing the aggregate workflow utilization:

"Overall Model Workflow

Multiple ML strategies were employed to cluster molecular profiles encoded as binary features in an unsupervised fashion. First, with the idea of capturing interdependencies across the features, we generated low dimensional ($l=32$) continuous representations of patient profiles using Autoencoders. Second, low dimensional representations were clustered using Gaussian Mixture Modeling optimizing over different number of components via BIC. Finally, running the Autoencoder-GMM steps over 100 runs with sub-sampling of input data and Keeping track of co-clustering of observations at each iteration, we generated a consensus-matrix representing the frequency of clustering observations in the same cluster. The generated consensus-matrix was further clustered using hierarchical-clustering with Ward's criteria to create the final cluster

assignments using Silhouette value to select the number of clusters¹³. One drawback of using a consensus matrix is that unseen data points cannot be readily clustered. In order to circumvent this issue, we used random forests to build a general classifier able to assign new molecular profiles into the clusters.”

2. A baseline model could be provided and compared with the model proposed. This will show the necessity of using this complex model involving autoencoder, GMM, hierarchical clustering, and RF.

Reply: We want to thank the reviewer for the excellent suggestion. We have generated a latent class analysis clustering as a baseline model to further justify the requirement of using a relatively more sensitive method. Specifically, we used BayesLCA similar to a previous study of our own to cluster binary mutation profiles in an unsupervised and Bayesian fashion (Awada et al. Blood (2021), PMID: 34075412). Indeed, applying this methodology resulted in lower number of clusters. We have included additional details in the main text as well. Specifically, we included:

“Furthermore, we have used a Bayesian latent class analysis as a baseline model for comparison using R package BayesLCA²⁶. As expected, clustering using the Bayesian approach resulted in more granular and lower resolution clusters (**Supplementary Figure 6A**) where substantial overlap with proposed risk groups was present (**Supplementary Figures 6B and 6C**). Nevertheless, the proposed Autoencoder based clustering was able to further stratify the BayesLCA based clusters.” **Supplementary Figure 6:**

3. With the current model evaluation results, I'm not convinced about the robustness. The similar number of clusters in each fold (in Line 151) is not enough to show the robustness. The calculation of the ARI is not clearly described in the manuscript or supplementary materials. According to the R script named by cluster_folds. R in the GitHub page provided by the author, ARI here is not fair to show the consistency of results.

Suppose the internal data is divided into 5 sets: set 1, 2, 3, 4, and 5.

Fold-1 whole training process involves set 2, 3, 4, and 5;

Fold-2 whole training process involves set 1, 3, 4, and 5.

When calculating the ARI between Fold-1 and Fold-2:

$ARI_{1,2} = ARI(\{\text{labels of set 2,3,4,5 predicted by Fold1}\}, \{\text{labels of set 2,3,4,5 predicted by Fold2}\})$

But set 3, 4, and 5 were training data in both Fold-1 and 2. The cross-validation evaluation results shouldn't contain shared training samples. Thus, I suggest the author to calculate the ARI between Fold-1 and Fold-2 by:

$ARI_{1,2} = ARI(\{\text{labels of set 1 predicted by Fold1}\}, \{\text{labels of set 1 predicted by Fold2}\})$

What is the statistical test for "no significant differences" in Line 156?

Reply: We want to thank the reviewer for the excellent comment. We acknowledge the fact that using the term "Fold" is misleading wherein individual Folds represent the 20% of the dataset set aside for training 5 distinct models. Furthermore, since there is no gold-standard for the clusters, we generated comparisons of 5 distinct models in an asymmetric fashion, where model 1 vs model 2 compares predictions done on 20% data, set aside as Fold 1. In contrast model 2 vs model 1 compares predictions done on 20% of data, set aside as Fold 2. Using this strategy, we compared predictions done on test set vs those done on the training set, given that the models (hence predictions) done on the training set for one of the compared models would be representative of a "gold-truth". Specifically, when comparing model 1 vs model 2, predictions of model 1 would act as gold-truth since model 1 was trained on fold 1, whereas model 2 predictions would act as test set predictions since model 2 has only been trained on fold 2.

Nevertheless, we have also generated ARI values for comparison done on "unseen" data for both models resulting in a symmetric matrix of ARI values. Specifically, when comparing models 1 and 2, the comparison is now done using 60% of dataset set aside for training models 3,4 and 5. Hence, neither of models 1 nor 2 have seen the test data. We included:

*"For the internal validation, we randomly selected training (80%, n=2870) and test (20%, n=718) sets for five-fold cross-validation to assess the fit of our model (See supplementary for details). Based on the highest Silhouette value in each fold, the majority of the folds (3 out of 5) showed 14 clusters as optimal number of total clusters, similar to the full cohort, suggesting robust strategy for our approach (**Supplementary Figure 5A**). Asymmetric and symmetric*

calculation of Adjusted-Rand Index (ARI) between the folds showed a minimum ARI of 0.85 (Supplementary Figures 5B and 5C).”

Supplementary Figure 5

A

B

C

4. Please provide summary figure or statistics for the claim in Line 171 “Blast percentages in MCs were consistent with the risk distribution of cases, and the median blast percentage was consistent with the composition of each MC.” The current results provided are not clear for this conclusion.

Reply: We thank the reviewer for this comment, we added **Supplementary Figure 9** which compares the median blast percent in each cluster. We showed that the composition of clusters (LR-MDS, HR-MDS, and sAML) is different (Table-2, p-value: <0.001). We included:

“Blast percentages in MCs were consistent with the risk distribution of cases, and the median blast percentage was consistent with the composition of each MC (**Table 2 and Supplementary Figure 9A**). For instance, while MC1 and MC13 had a median blast percentage of > 10%, MC2 and MC4 had a median of <5%, consistent with the enrichment of

early-stage (LR-MDS) cases within the latter MCs. Overall, MC1 and MC13 had significantly higher odds for $\geq 20\%$ BM blast percentage while MC2 and MC4 had higher odds for $< 20\%$ BM blast percentage (Supplementary Figure 9B). ”

5. Besides Supplementary Figure 11 and the example in the very low IPSS-R group, please provide more evidence to rule out the confounding factor effect of IPSS-R in studying the overall survival differences among MCs, since it was mentioned in Line 167 that “the distribution of different revised international prognostic scoring system (IPSS-R) risk groups among our MCs were distinct and heterogenous.” This is essential in determining the novelty of the findings in OS differences.

Reply: We want to thank the reviewer for the excellent comment. To rule out the possible confounding effects of IPSS-R and/or similarly IPSS-M in studying the overall survival differences among molecular clusters and risk groups, we generated CoxPH model accounting for IPSS-M risk stratification. Both IPSS-M and the proposed risk groups showed significant difference in HR compared to the baseline (Group 1/Low Risk, IPSS-M Very Low and Female). We observed a time-dependence in IPSS-M groups and age variable as well

where an arbitrary cutoff of 30 months to account for proportionality assumption is included. We included in the text as follows:

*“To rule out possible confounding effect of IPSS-M and IPSS-R in estimating the OS differences among our risk groups, we generated a Cox PH model including IPSS-M/IPSS-R and clinical variables (**Supplementary Table 10 and Supplementary Table 11**). As expected, both models continued to show significant association with OS.”*

Supplementary Table 10 Hazard-Risk (HR) Coefficients and associated CI and P-values for our risk groups adjusted for IPSS-R and IPSS-M

Factor	Training Set (HR, CI, P-Value)
Sex (Male)	1.35 (1.13-1.61, 6.36E-4)
log(BM+1.0)	0.84 (0.73-0.97, 1.92E-2)
log(PLT+1.0)	0.82 (0.75-0.91, 1.81E-4)
HB	0.93 (0.88-0.99, 2.31E-2)
Risk Int-Low	1.27 (0.98-1.64, 6.24E-2)
Risk Int-High	1.47 (1.11-1.95, 6.72E-3)
Risk High	1.53 (1.02-2.29, 3.54E-2)
Risk Poor	2.48 (1.66-3.69, 7.90E-6)
Age (Time 1)	1.01 (1.008-1.02, 1.76E-4)
Age (Time 2)	1.06 (1.044-1.07, 8.90E-13)
IPSSM Low (Time 1)	1.31 (0.84-2.02, 2.22E-1)
IPSSM Low (Time 2)	1.23 (0.81 – 1.86, 3.29E-1)
IPSSM Moderate Low (Time 1)	1.74 (1.03 – 2.91, 3.50E-2)
IPSSM Moderate Low (Time 2)	1.55 (0.89 – 2.70, 1.15E-1)
IPSSM Moderate High (Time 1)	2.82 (1.67 – 4.75, 9.17E-5)
IPSSM Moderate High (Time 2)	1.29 (0.71 – 2.33, 3.97E-1)
IPSSM High (Time 2)	4.03 (2.33 – 6.98, 6.29E-7)
IPSSM High (Time 2)	1.77 (0.93 – 3.35, 7.70E-2)

Minor:

1. Figure 2 is rarely discussed in the manuscript. Should the author attach more writing to the results if it is essential, or put it in supplementary figures and take other plots to the main figure?

Reply: We thank the reviewer for the comment. Figure 2 illustrates the molecular features in each cluster, and we cited it more often in the main article (lines 195, 207, 217) and highlighted these changes.

2. The color palette consistency

In Figure 1, please make sure each label has only one color. E.g., A and C have different label colors for LR-MDS, HR-MDS, and sAML.

The five risk group labels are inconsistent between Supplementary Figure 10A and others.

Reply: We thank the reviewer for the comment. We changed the labels in Figure 1C and Supplementary Figure 10A to ensure consistency. Furthermore, we reviewed the labels for all other figures to ensure homogenous colors.

REVIEWERS' COMMENTS

Reviewer #1 (Remarks to the Author):

The authors have done a very thorough revision and addressed all of my points. They added a lot of new analyses and also improved online tool. There are no any additional remarks from my side.

Reviewer #2 (Remarks to the Author):

All my concerns have been addressed.